# Cardiac arrhythmias in hospitalized patients with COVID-19: A prospective observational study in the western United States

**Jae Hyung Cho, Ali Namazi, Richard Shelton, Archana Ramireddy, Ashkan Ehdaie, Michael Shehata, Xunzhang Wang, Eduardo Marbán, Sumeet S. Chugh, Eugenio Cingolani**📖 *

Smidt Heart Institute, Cedars-Sinai Medical Center, Los Angeles, California, United States of America

* Eugenio.Cingolani@csmc.edu

**Data Availability Statement:** All relevant data are within the manuscript.

## Abstract

Arrhythmias have been reported frequently in COVID-19 patients, but the incidence and nature have not been well characterized. Patients admitted with COVID-19 and monitored by telemetry were prospectively enrolled in the study. Baseline characteristics, hospital course, treatment and complications were collected from the patients' medical records. Telemetry was monitored to detect the incidence of cardiac arrhythmias. The incidence and types of cardiac arrhythmias were analyzed and compared between survivors and non-survivors. Among 143 patients admitted with telemetry monitoring, overall in-hospital mortality was 25.2% (36/143 patients) during the period of observation (mean follow-up 23.7 days). Survivors were less tachycardic on initial presentation (heart rate 90.6 ± 19.6 vs. 99.3 ± 23.1 bpm, p = 0.030) and had lower troponin (peak troponin 0.03 vs. 0.18 ng/ml. p = 0.004), C-reactive protein (peak C-reactive protein 97 vs. 181 mg/dl, p = 0.029), and interleukin-6 levels (peak interleukin-6 30 vs. 246 pg/ml, p = 0.003). Sinus tachycardia, the most common arrhythmia (detected in 39.9% [57/143] of patients), occurred more frequently in non-survivors (58.3% vs. 33.6% in survivors, p = 0.009). Premature ventricular complexes occurred in 28.7% (41/143), and non-sustained ventricular tachycardia in 15.4% (22/143) of patients, with no difference between survivors and non-survivors. Sustained ventricular tachycardia and ventricular fibrillation were not frequent (seen only in 1.4% and 0.7% of patients, respectively). Contrary to reports from other regions, overall mortality was higher and ventricular arrhythmias were infrequent in this hospitalized and monitored COVID-19 population. Either disease or management-related factors could explain this divergence of clinical outcomes, and should be urgently investigated.

## Introduction

The 2019 novel coronavirus disease (COVID-19) caused by severe acute respiratory syndrome coronavirus 2 (SARS-CoV2) resulted in a worldwide, rapidly spreading and high mortality pandemic. Although COVID-19 is a respiratory illness, acute cardiac injury defined by elevated troponin, electrocardiographic or echocardiographic abnormalities was reported in 12%

**Funding:** This study was funded by the National Institutes of Health (RO1 HL135866 to EC and EM) and the Peer-Reviewed Medical Research Program of the US Department of Defense (PR150620 to EM). The funders had no role in study design, data collection and analysis, decision to publish, or preparation of the manuscript.

**Competing interests:** The authors have declared that no competing interests exist.

(5/41 patients) of initial patients admitted to the hospital in Wuhan, China [1]. A subsequent study also confirmed that cardiac injury occurred in 19.7% (82/416 patients) of hospitalized patients and was further associated with increased mortality (51.2% with cardiac injury vs. 4.5% without cardiac injury) [2]. Cause of death analysis in COVID-19 patients revealed that approximately 40% of deaths were due to myocardial damage and/or heart failure [3]. A recent study of 138 hospitalized COVID-19 patients in Wuhan, China revealed that cardiac arrhythmias were common complications occurring in 16.7% (23/138) of patients but details about the arrhythmias were not reported [4]. Another study in China revealed that ventricular tachycardia (VT) or ventricular fibrillation (VF) occurred in 17.3% (9/52 patients) of hospitalized COVID-19 patients with elevated troponin [5]. The usual conjecture is that COVID-19 can cause cardiac injury via myocarditis leading to heart failure, thus increasing the risk of VT or VF. Increased sympathetic tone and cytokine storm have also been postulated as a cause of ventricular arrhythmias (VA) [6]. Although VT or VF were frequent complications in patients with COVID-19 in China, the detailed incidence and characteristics of cardiac arrhythmias in other parts of the world are unknown. We conducted a prospective observational study to analyze the incidence and predictors of cardiac arrhythmias in patients admitted to the hospital with COVID-19 in an academic institution in the Western United States.

## Materials and methods

### Ethics statement and study approval

This study was presented to the Institutional Review Board of Cedars-Sinai Medical Center and approval was obtained before the initiation (Study 00000638). The nature of this study is a prospective observational case series of patients with COVID-19 who were admitted and monitored by telemetry. Informed consents were waived due to the nature of this study (prospective medical records review).

### Identification of patients

We prospectively identified patients with COVID-19 who were admitted to the hospital (either intensive care unit or ward) from March 15, 2020 to April 30, 2020 (Fig 1). Patients who were monitored by continuous cardiac telemetry were included in this study. Patients who were not monitored were excluded. Demographic and clinical data were reviewed from the patients' medical records. Review of cardiac telemetry was performed from hospital admission until discharge or death.

### Baseline characteristics

Baseline characteristics (age, sex, body mass index, hypertension, diabetes mellitus, hyperlipidemia, coronary artery disease, chronic kidney disease, atrial fibrillation, heart failure with reduced ejection fraction or preserved ejection fraction) were collected from the patients' medical records. Baseline ECG characteristics were measured on the initial admission ECG (PR interval, QRS width, QT and QTc intervals), and presenting rhythms (normal sinus rhythm, sinus tachycardia, sinus bradycardia, atrial fibrillation or flutter, first degree AV block, paced rhythm, premature atrial or ventricular complexes) were recorded. These characteristics were compared between survivors and non-survivors.

### Hospital course, treatment and complications

Patients were prospectively monitored for the development of major complications (acute respiratory distress syndrome, septic shock requiring inotropes or vasopressors, hemodialysis,

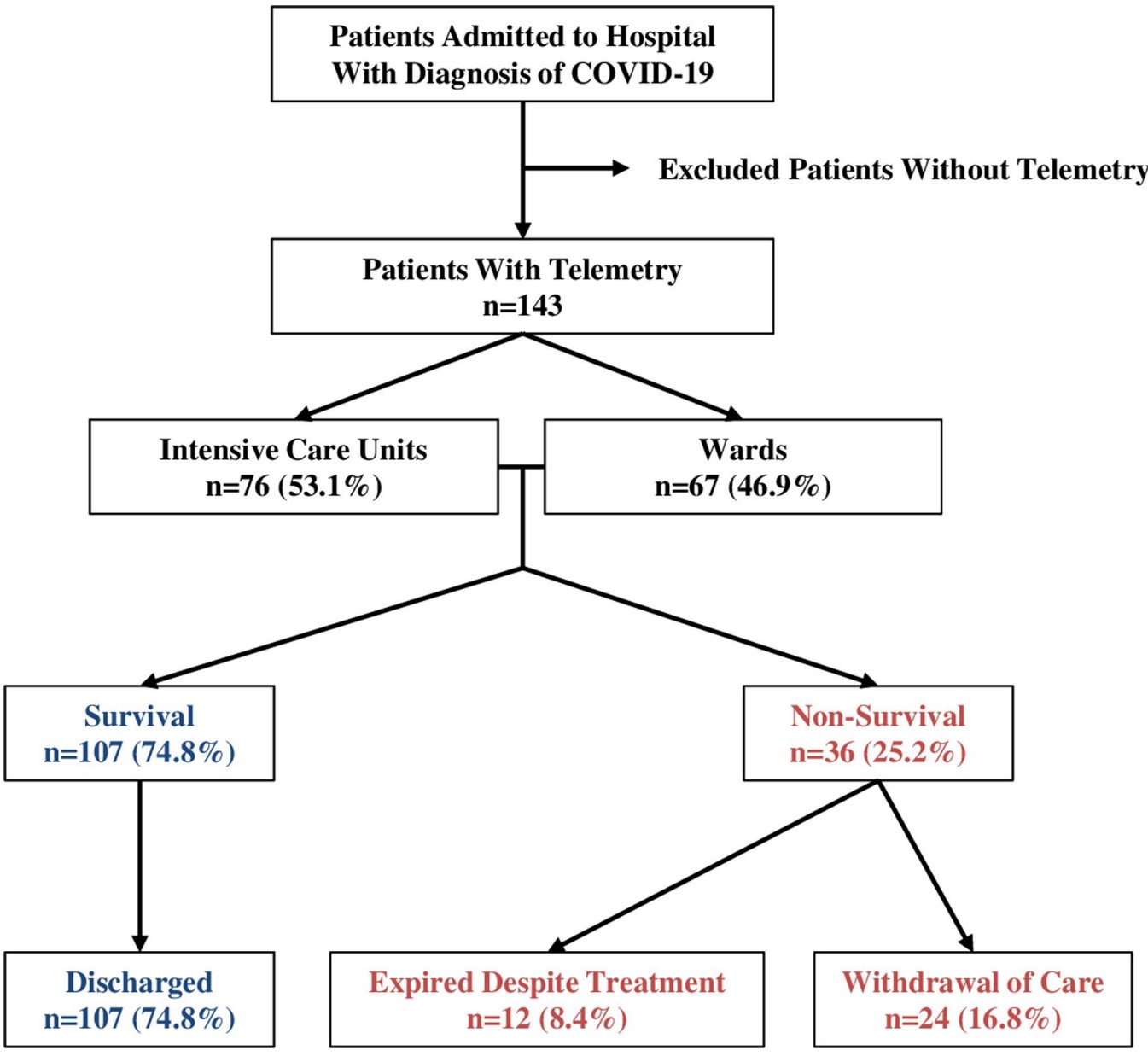

**Fig 1. Patient identification and discharge summary.**

venous thromboembolism, arterial thrombosis and extracorporeal membrane oxygenation). Transthoracic echocardiograms were reviewed to investigate the development of left or right ventricular dysfunction. Relevant laboratory data (peak troponin, B-type natriuretic peptide, C-reactive protein, and interleukin-6) were also reviewed. Treatment modalities (antibiotics, anti-interleukin-6 monoclonal antibody and anti-viral agents) were also recorded. These were also compared between survivors and non-survivors.

### Telemetry monitoring

Telemetry recordings were monitored during the hospitalization. Development of cardiac arrhythmias was investigated (sinus tachycardia, sinus bradycardia, premature atrial or

ventricular complexes, supraventricular tachycardia, atrial fibrillation, new-onset atrial fibrilla-tion, atrial flutter, VT, VF, complete AV block, and sinus arrest). Non-sustained VT was defined as less than 30 seconds and sustained VT as more than 30 seconds.

### In-hospital cardiac arrest

Patients were also monitored for in-hospital cardiac arrest and events were analyzed from the medical records. Initial rhythms (VT/VF, asystole or pulseless electrical activity [PEA]) and the outcomes of cardiac arrest were recorded and analyzed.

### Statistical analyses

SPSS Statistics version 24 was used to perform the statistical analysis. Baseline patient charac-teristics are presented as numbers and percentages for categorical variables, mean ± standard deviation for normally distributed continuous variables and median (interquartile range) for non-normally distributed continuous variables. Comparisons of categorical variables were per-formed using Pearson's chi-square test (expected value more than 5) or Fisher's exact test (expected value less than 5) and comparisons of continuous variables were performed using Student t-test (normally distributed variables) or Mann-Whitney test (non-normally distrib-uted variables). Results were considered significant at p values < 0.05.

## Results

### Patient population, discharge and mortality

A total of 143 patients diagnosed with COVID-19 were admitted to hospital with telemetry monitoring (Fig 1). Out of 143 patients, 76 patients were admitted to intensive care units (53.1%) and 67 patients to telemetry ward (46.9%) (Fig 1). Overall in-hospital mortality was 25.2% (36/143), with 12 patients expiring despite aggressive treatment (8.4%) and 24 patients after withholding aggressive treatment (comfort care or hospice, 16.8%). The rest of the patients (107 patients) were discharged to either home or a skilled nursing facility with an overall survival rate of 74.8%.

### Baseline characteristics

Baseline characteristics of the 143 patients with COVID-19 are described in Table 1. Most of the baseline characteristics were not statistically different between survivors and non-survivors except for age and body mass index. The survivors were significantly younger and had higher body mass index than the non-survivors. There were no significant differences in the present-ing rhythms on the initial ECGs. However, initial heart rate was significantly higher in non-survivors compared to survivors. Intervals on initial ECGs were not statistically different between survivors and non-survivors.

### Hospital course, treatment and complications

There were no significant differences in the rate of complications between survivors and non-survivors (Table 2). Transthoracic echocardiograms were done on 55/143 patients and there were no significant differences in left ventricular ejection fraction and right ventricular func-tion. Peak troponin, peak C-reactive protein and peak interleukin-6 levels were significantly higher in non-survivors compared to survivors. B-type natriuretic peptides levels were not sta-tistically different between survivors and non-survivors. There were no significant differences in the modalities of treatments between the two groups.

**Table 1. Baseline characteristics of COVID-19 patients.**

|  | Total (n = 143) | Survivors (n = 107) | Non-survivors (n = 36) | P value |
|---|---|---|---|---|
| Age (mean ± SD) | 70.3 ± 17.3 | 68.5 ± 17.2 | 75.8 ± 16.8 | 0.027 |
| Sex (male) | 88 (61.5%) | 68 (63.6%) | 20 (55.6%) | 0.394 |
| Body mass index (mean ± SD) | 27.2 ± 8.0 | 28.2 ± 8.4 | 24.2 ± 6.0 | 0.008 |
| Hypertension | 79 (55.2%) | 63 (58.9%) | 16 (44.4%) | 0.132 |
| Diabetes mellitus | 50 (35.0%) | 41 (38.3%) | 9 (25.0%) | 0.147 |
| Hyperlipidemia | 59 (41.3%) | 45 (42.1%) | 14 (38.9%) | 0.738 |
| Coronary artery disease | 27 (18.9%) | 23 (21.5%) | 4 (11.1%) | 0.221 |
| Chronic kidney disease | 29 (20.3%) | 19 (17.8%) | 10 (27.8%) | 0.196 |
| Atrial fibrillation | 18 (12.6%) | 13 (12.1%) | 5 (13.9%) | 0.776 |
| Heart failure with reduced EF | 10 (7.0%) | 8 (7.5%) | 2 (5.6%) | 1.000 |
| Heart failure with preserved EF | 16 (11.2%) | 14 (13.1%) | 2 (5.6%) | 0.359 |
| Baseline ECG |  |  |  |  |
| Normal sinus rhythm | 80 (55.9%) | 62 (58.5%) | 18 (50.0%) | 0.375 |
| Sinus tachycardia | 40 (28.0%) | 26 (24.5%) | 14 (38.9%) | 0.098 |
| Sinus bradycardia | 5 (3.5%) | 4 (3.8%) | 1 (2.8%) | 1.000 |
| Atrial fibrillation | 8 (5.6%) | 5 (4.7%) | 3 (8.3%) | 0.418 |
| Atrial flutter | 5 (3.5%) | 5 (4.7%) | 0 (0.0%) | 0.330 |
| First degree AV block | 7 (4.9%) | 6 (5.7%) | 1 (2.8%) | 0.679 |
| Paced | 2 (1.4%) | 2 (1.9%) | 0 (0.0%) | 1.000 |
| Premature atrial complex | 4 (2.8%) | 2 (1.9%) | 2 (5.9%) | 0.266 |
| Premature ventricular complex | 3 (2.1%) | 2 (1.9%) | 1 (2.8%) | 1.000 |
| HR (bpm, mean ± SD) | 92.8 ± 20.8 | 90.6 ± 19.6 | 99.3 ± 23.1 | 0.030 |
| PR interval (ms, mean ± SD) | 157.4 ± 30.0 | 158.7 ± 30.5 | 153.7 ± 28.6 | 0.415 |
| QRS width (ms, mean ± SD) | 94.0 ± 21.3 | 93.4 ± 20.4 | 95.9 ± 24.1 | 0.537 |
| QT interval (ms, mean ± SD) | 370.4 ± 51.5 | 372.8 ± 51.6 | 363.6 ± 51.1 | 0.359 |
| QTc interval (ms, mean ± SD) | 451.8 ± 31.9 | 449.8 ± 28.8 | 457.8 ± 39.5 | 0.197 |

## Telemetry monitoring of COVID-19 patients

The most common cardiac arrhythmias in patients with COVID-19 were sinus tachycardia (57/143 = 39.9%) followed by premature ventricular complexes (41/143 = 28.7%), non-sustained VT (22/143 = 15.4%) and atrial fibrillation (17/143 = 11.9%) (Table 3). Sinus tachycardia was associated with increased mortality (58.3% in non-survivors vs. 33.6% in survivors, p = 0.009). The average number of beats in non-sustained VT was 7.3 ± 4.7 beats. The incidence of non-sustained VT was not significantly different between the two groups. New-onset atrial fibrillation was not different between survivors and non-survivors. Sustained VT and VF were not frequent and occurred only in 2 patients (1.4%) and 1 patient (0.7%), respectively (Fig 2). Similarly, complete AV block occurred only in 2 patients (1.4%) and sinus arrest in 1 patient 0.7%) (Fig 2).

## In-hospital cardiac arrest

A total of 13 patients developed cardiac arrest during hospitalization (9.1%) (Fig 3). Three patients were full code and received cardiopulmonary resuscitation (2.1%). Initial rhythms were VT, asystole and PEA for each patient, and only the patient with VT survived. Ten patients were do-not-resuscitate status and expired (7.0%) without cardiopulmonary resuscitation. Cardiac rhythms at the time of death were VT/VF for one patient (0.7%), asystole for 7 patients (4.9%) and PEA for 2 patients (1.4%). Twenty-four patients withdrew care (comfort care or hospice) and expired without rhythm monitoring (16.8%).

**Table 2. Hospital course, complications and treatment of COVID-19 patients.**

| | Total (n = 143) | Survivors (n = 107) | Non-survivors (n = 36) | P value |
|---|---|---|---|---|
| **Acute respiratory distress syndrome** | 50 (35.0%) | 38 (35.5%) | 12 (33.3%) | 0.812 |
| **Septic shock requiring vasopressors** | 24 (16.8%) | 17 (15.9%) | 7 (19.4%) | 0.621 |
| **Hemodialysis** | 18 (12.6%) | 12 (11.2%) | 6 (16.7%) | 0.394 |
| **Venous thromboembolism** | 18 (12.6%) | 16 (15.0%) | 2 (5.6%) | 0.243 |
| **Arterial thrombosis** | 4 (2.8%) | 3 (2.8%) | 1 (2.8%) | 1.000 |
| **Extracorporeal membrane oxygenation** | 3 (2.1%) | 2 (1.9%) | 1 (2.8%) | 1.000 |
| **Echocardiogram** | (n = 55) | (n = 46) | (n = 9) | |
| Left ventricular ejection fraction (%) | 57.8 ± 11.3 | 58.2 ± 10.0 | 56.8 ± 14.9 | 0.696 |
| Right ventricular dysfunction | 18 (32.7%) | 12 (29.3%) | 6 (42.9%) | 0.349 |
| **Labs** | | | | |
| Peak Troponin (ng/ml), median (IQR) | 0.04 (0.18) | 0.03 (0.11) | 0.18 (0.97) | 0.004 |
| Peak BNP (pg/ml), median (IQR) | 117 (293) | 114 (249) | 150 (380) | 0.256 |
| Peak CRP (mg/dl), median (IQR) | 109 (154) | 97 (141) | 181 (154) | 0.029 |
| Peak Interleukin-6 (pg/ml), median (IQR) | 42 (199) | 30 (131) | 246 (664) | 0.003 |
| **Treatment** | | | | |
| Azithromycin | 85 (59.4%) | 64 (59.8%) | 21 (58.3%) | 0.876 |
| Hydroxychloroquine | 90 (62.9%) | 70 (65.4%) | 20 (55.6%) | 0.289 |
| Tocilizumab | 56 (39.2%) | 44 (41.1%) | 12 (33.3%) | 0.408 |
| Lopinavir/Ritonavir | 3 (2.1%) | 3 (2.8%) | 0 (0.0%) | 0.572 |
| Remdesivir | 13 (9.1%) | 12 (11.2%) | 1 (2.8%) | 0.185 |

## Comparison between COVID-19 patients with normal vs. elevated troponin

Troponin levels were measured for 141 patients during their hospitalization. As compared to patients with a normal troponin level (≤0.04 ng/ml), patients with elevated troponin (>0.04 ng/ml) demonstrated a higher in-hospital mortality (Table 4). There were no differences in monitored rhythms during hospitalization except for non-sustained VT and new-onset atrial fibrillation. The incidence of non-sustained VT was significantly higher in patients with

**Table 3. Telemetry monitoring of COVID-19 patients.**

| | Total (n = 143) | Survivors (n = 107) | Non-survivors (n = 36) | P value |
|---|---|---|---|---|
| **Sinus tachycardia** | 57 (39.9%) | 36 (33.6%) | 21 (58.3%) | 0.009 |
| **Sinus bradycardia** | 7 (4.9%) | 6 (5.6%) | 1 (2.8%) | 0.680 |
| **Premature atrial complex** | 11 (7.7%) | 7 (6.5%) | 4 (11.1%) | 0.469 |
| **Premature ventricular complex** | 41 (28.7%) | 32 (29.9%) | 9 (25.0%) | 0.573 |
| **Supraventricular tachycardia** | 8 (5.6%) | 4 (3.7%) | 4 (11.1%) | 0.110 |
| **Atrial fibrillation** | 17 (11.9%) | 11 (10.3%) | 6 (16.7%) | 0.543 |
| **New-onset atrial fibrillation** | 8 (5.6%) | 4 (3.7%) | 4 (11.1%) | 0.110 |
| **Atrial flutter** | 3 (2.1%) | 3 (2.8%) | 0 (0.0%) | 0.572 |
| **Ventricular tachycardia** | | | | |
| Non-sustained | 22 (15.4%) | 15 (14.0%) | 7 (19.4%) | 0.435 |
| Sustained | 2 (1.4%) | 0 (0.0%) | 2 (5.6%) | 0.062 |
| **Ventricular fibrillation** | 1 (0.7%) | 0 (0.0%) | 1 (2.8%) | 0.252 |
| **Complete AV block** | 2 (1.4%) | 1 (0.9%) | 1 (2.8%) | 0.441 |
| **Sinus arrest** | 1 (0.7%) | 0 (0.0%) | 1 (2.8%) | 0.252 |

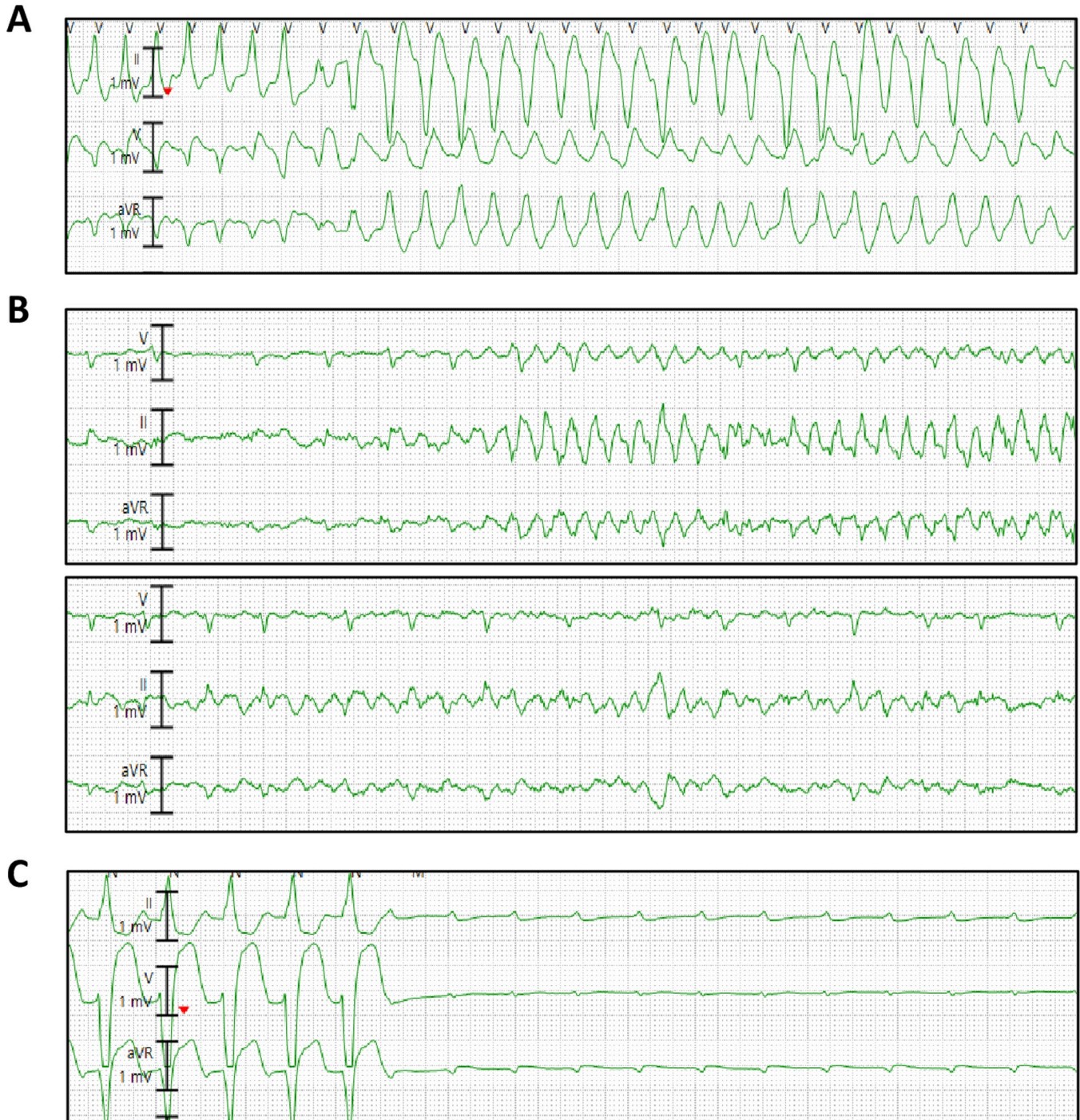

**Fig 2. Malignant arrhythmias in three patients with COVID-19.** A. Sustained polymorphic ventricular tachycardia. B. Polymorphic ventricular tachycardia degenerating into ventricular fibrillation. C. Complete AV block.

elevated troponin as compared to normal troponin. New-onset atrial fibrillation was more common in patients with elevated troponin than normal troponin. Importantly, all life-threatening arrhythmias (sustained VT, VF, complete AV block and sinus arrest) occurred only in patients with elevated troponin.

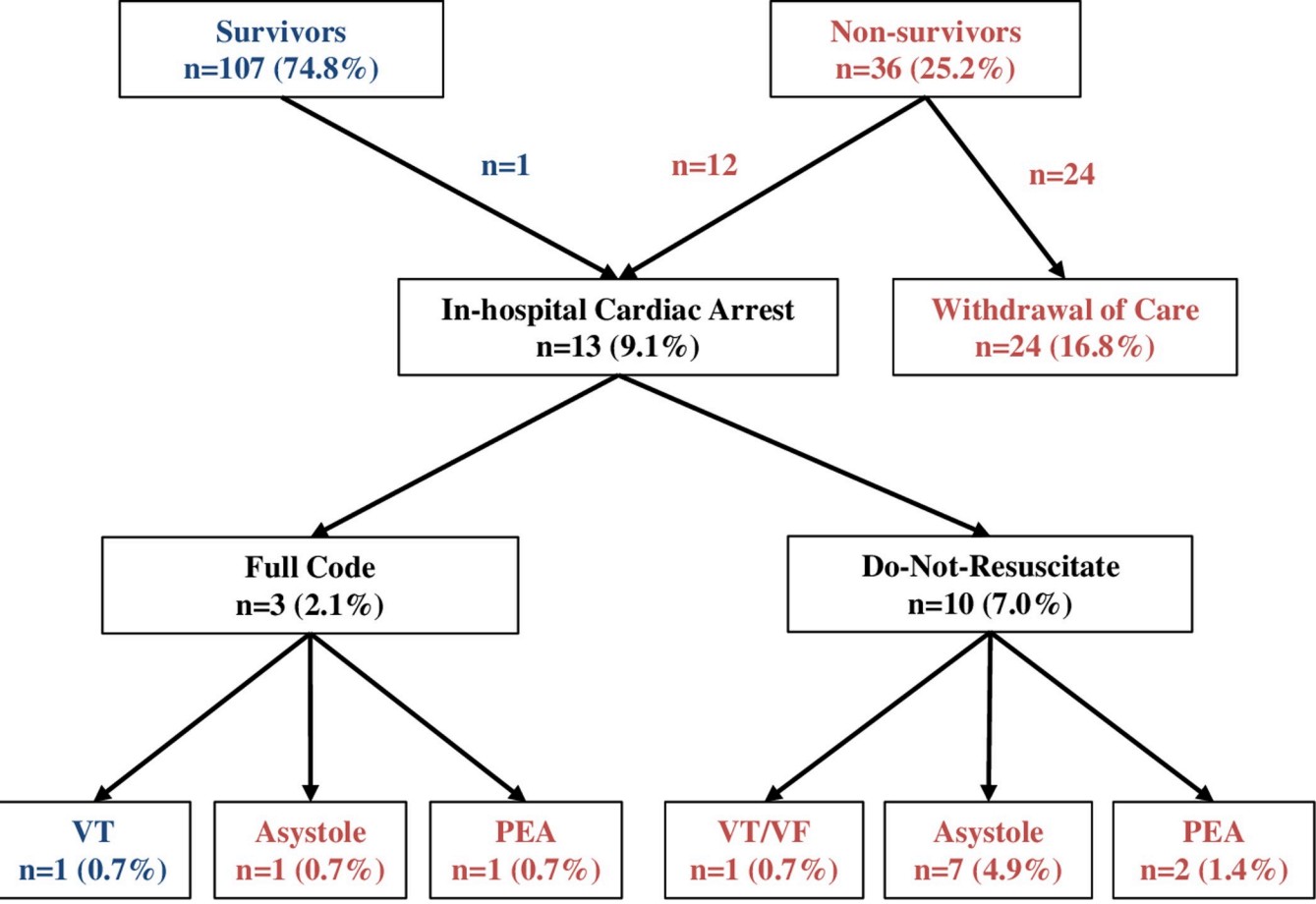

**Fig 3. In-hospital cardiac arrest of COVID-19 patients.**

**Table 4. Incidence of cardiac arrhythmias in COVID-19 patients with normal vs. elevated troponin.**

|  | Total (n = 141) | Normal troponin (n = 72) | Elevated troponin (n = 69) | P value |
|---|---|---|---|---|
| Mortality | 30 (21.2%) | 12 (16.7%) | 24 (34.8%) | 0.014 |
| Sinus tachycardia | 56 (39.7%) | 28 (38.9%) | 28 (40.6%) | 0.837 |
| Sinus bradycardia | 7 (5.0%) | 4 (5.6%) | 3 (4.3%) | 1.000 |
| Premature atrial complex | 11 (7.8%) | 3 (4.2%) | 8 (11.6%) | 0.124 |
| Premature ventricular complex | 40 (28.4%) | 17 (23.6%) | 23 (33.3%) | 0.200 |
| Supraventricular tachycardia | 8 (5.7%) | 3 (4.2%) | 5 (7.2%) | 0.487 |
| Atrial fibrillation | 17 (12.1%) | 5 (6.9%) | 12 (17.4%) | 0.072 |
| New-onset atrial fibrillation | 8 (5.7%) | 1 (1.4%) | 7 (10.1%) | 0.031 |
| Atrial flutter | 3 (2.1%) | 2 (2.8%) | 1 (1.4%) | 1.000 |
| Ventricular tachycardia |  |  |  |  |
| Non-sustained | 22 (15.6%) | 5 (6.9%) | 17 (24.6%) | 0.005 |
| Sustained | 2 (1.4%) | 0 (0.0%) | 2 (2.9%) | 0.238 |
| Ventricular fibrillation | 1 (0.7%) | 0 (0.0%) | 1 (1.4%) | 0.489 |
| Complete AV block | 2 (1.4%) | 0 (0.0%) | 2 (2.9%) | 0.238 |
| Sinus arrest | 1 (0.7%) | 0 (0.0%) | 1 (1.4%) | 0.489 |

## Discussion

In this prospective observational study of hospitalized and monitored patients with COVID-19, sinus tachycardia was the most common rhythm disorder and its presence was associated with higher in-hospital mortality. Sustained VT or VF were rare occurring only in 1.4% and 0.7% of patients, respectively. Our findings are in contrast to the previous observations that malignant VA are commonly seen in hospitalized patients with COVID-19 [4, 5].

A recent study of 138 hospitalized COVID-19 patients in Wuhan, China revealed that cardiac arrhythmias were common complications occurring in 16.7% (23/138) of patients, but the details of VA were not described [4]. A subsequent study in China revealed that VT or VF occurred in 5.9% (11/187 patients) of hospitalized COVID-19 patients, and the incidence was even higher (17.3%, 9/52 patients) in those with elevated troponin [5]. Our study showed that malignant VA were infrequent in patients with COVID-19 admitted to our institution. Our findings are quite different from those reported from Wuhan, China. Several potential factors could explain the lower incidence of malignant arrhythmias observed at our center. Firstly, our demographic was different. Our study population (mean age 70 years old) was approximately 10 years older than the population reported in China (mean age 56 and 63 years old) [2, 4]. One possibility is that older patients might have more respiratory system-related complications rather than cardiac injury with malignant VA. Secondly, the admission criteria may have been different in different centers. Compared to Wuhan, China where only severe cases were admitted for monitoring and treatment (mortality of patients with cardiac injury 51.2%) [2], in Los Angeles, USA less severe cases may have been admitted (mortality of patients with cardiac injury 34.8%). The degree of illness may impact in-hospital cardiovascular complications, specifically cardiac arrhythmias. Third, differences in medical management may play a role. As treatment algorithms evolved around the world from the early experiences in China, combination therapy with antiviral and anti-inflammatory agents may have had impact on the development of cardiac injury and in-hospital cardiac arrhythmias. Lastly, disease-related factors such as viral strain might have impacted the differences in the development of cardiac arrhythmias. Another study from the US, the University of Pennsylvania, reported that cardiac arrests were not frequent (25/700 patients), mostly due to either PEA or asystole [7]. These data are consistent with our findings and support that disease-related factors might have played a role.

A recent study from China investigating in-hospital cardiac arrests in COVID-19 patients revealed that only 5.9% of cardiac arrests were due to VT or VF [8]. Most of the in-hospital cardiac arrests were caused by asystole (89.7%) or PEA (4.4%) [8]. The mechanisms of VA in patients with COVID-19 are currently not known. VA are frequent complications of patients with acute myocarditis [9]. In patients with active myocarditis, sustained VT was found in 24% (30/123 patients) of patients, and VF was present in 7% (8/123 patients) [10]. Most cases of acute myocarditis are caused by viruses [11], however, the prevalence of VA in patients with viral myocarditis is not known. The mechanisms of VA caused by viral myocarditis are likely multifactorial including inflammation, microvascular ischemia, gap junction dysfunction, and scar formation [12]. Cardiac arrhythmias can potentially occur in patients with COVID-19 as a result of myocarditis caused by direct invasion of the virus into the heart, exaggerated inflammatory response or cytokine storm [6]. Elevated cardiac enzymes are possibly due to viral myocarditis. However, without cardiac magnetic resonance imaging (MRI) and/or endomyocardial biopsy, a diagnosis of viral myocarditis may be speculative. Many alternative causes of elevated cardiac enzymes including coronary artery plaque rupture, thromboembolism, and heart failure exist in cases of in-hospital illness. Without cardiac MRI, endomyocardial biopsy and coronary angiogram, the link between a speculative diagnosis of viral myocarditis and VA

is weak. Exaggerated inflammatory response and cytokine storm are evident from increased inflammatory markers and elevated interleukin-6 levels in our study. The association between increased inflammatory markers and development of VA were not studied due to the low prevalence of malignant arrhythmias in our study population.

Sinus tachycardia was the most common arrhythmia in patients with COVID-19. Sinus tachycardia is likely multifactorial caused by fever, dyspnea, hypotension or shock, and can be an inflammatory response against SARS-CoV2. Our findings suggest that it serves as a significant prognostic factor in our COVID-19 patients.

Our study demonstrated an in-hospital mortality of 25.2%, higher than the reported mortality of ~10% in China [1, 2, 4]. This discrepancy may be explained by the older population in our study as compared to prior studies from China. Population differences in co-morbid conditions may also play a role. Consistently with the previous studies, elevated troponin levels were associated with mortality in our patient population. There is a possibility that the patients with coronary artery disease might have avoided seeing a doctor or coming to a hospital although they developed cardiac symptoms, due to the concerns of exposure to COVID-19 [13]. Such patients might have come to hospital eventually when they contracted COVID-19 and this could have led to high levels of troponin and high mortality. Non-sustained VT and new-onset atrial fibrillation were more common in patients with elevated troponin compared to normal troponin. Notably, serious arrhythmias (sustained VT, VF, complete AV block and sinus arrest) occurred only in patients with elevated troponin.

The conclusions of the present study must be tempered by several limitations. First, the observational nature of this study introduces selection bias related to admitted and monitored patients with COVID-19. The incidence of cardiac arrhythmias in hospitalized but non-monitored patients and patients with mild illness at home is unknown. Therefore, the overall incidence of malignant VA would likely be lower than the one reported here. Comparisons between COVID-19 patients who were hospitalized vs. non-hospitalized and even with healthy controls would have potentiated our findings. Secondly, the low number of patients admitted and monitored may not adequately represent the true incidence of malignant arrhythmias in the COVID-19 population. Third, this was a single center observational study and the incidence of malignant VA might be different in other parts of the country. Finally, due to the low incidence of serious VA in our patient population, there was no significant association between malignant VA and mortality. Thus, a large-scale surveillance of cardiac arrhythmias in patients with COVID-19 is needed.

## Acknowledgments

We would like to appreciate Drs. Siddharth Singh, Yuri Matusov and Oren Friedman for their help in getting transthoracic echocardiogram for COVID-19 patients.

## Author Contributions

**Conceptualization:** Jae Hyung Cho, Ali Namazi, Richard Shelton, Archana Ramireddy, Ashkan Ehdaie, Michael Shehata, Xunzhang Wang, Eduardo Marbán, Sumeet S. Chugh, Eugenio Cingolani.

**Data curation:** Jae Hyung Cho, Ali Namazi, Richard Shelton, Archana Ramireddy.

**Formal analysis:** Jae Hyung Cho.

**Investigation:** Jae Hyung Cho, Xunzhang Wang, Sumeet S. Chugh, Eugenio Cingolani.

**Methodology:** Jae Hyung Cho, Ali Namazi, Richard Shelton, Archana Ramireddy, Ashkan Ehdaie, Michael Shehata, Eduardo Marbán, Sumeet S. Chugh, Eugenio Cingolani.

**Supervision:** Michael Shehata, Xunzhang Wang, Sumeet S. Chugh, Eugenio Cingolani.

**Writing – original draft:** Jae Hyung Cho, Eugenio Cingolani.

**Writing – review & editing:** Jae Hyung Cho, Ashkan Ehdaie, Michael Shehata, Eduardo Marbán, Sumeet S. Chugh, Eugenio Cingolani.

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
