## [Decision Letter · Decision Letter 0]

5 Nov 2020

PONE-D-20-25283

Cardiac Arrhythmias in Hospitalized Patients with COVID-19: A Prospective Observational Study in the Western United States

PLOS ONE

Dear Dr. Cingolani,

Thank you for submitting your manuscript to PLOS ONE. After careful consideration, we feel that it has merit but does not fully meet PLOS ONE’s publication criteria as it currently stands. Therefore, we invite you to submit a revised version of the manuscript that addresses the points raised during the review process.

Please address minor comments from the Reviewer.

We look forward to receiving your revised manuscript.

Kind regards,

Elena G. Tolkacheva, PhD

Academic Editor

PLOS ONE

Journal Requirements:

Additional Editor Comments (if provided):

Reviewers' comments:

Reviewer's Responses to Questions

**Comments to the Author**

1. Is the manuscript technically sound, and do the data support the conclusions?

Reviewer #1: Yes

2. Has the statistical analysis been performed appropriately and rigorously? 

Reviewer #1: Yes

3. Have the authors made all data underlying the findings in their manuscript fully available?

Reviewer #1: Yes

4. Is the manuscript presented in an intelligible fashion and written in standard English?

Reviewer #1: Yes

5. Review Comments to the Author

Reviewer #1: This is an interesting and potentially important manuscript which examines caridac arrhythmias in patients who are hospitalized because of Covid 19. The study was limited to patients who were placed on telemetry. A total of 143 patients were enrolled, all of them from a single site. The overall mortality was 25.2%. The major conclusion from the telemetry was that the survivors were significantly less tachycardic than the patients who did not survive (mean heart rate 90.6 BMP vs. 99.3 BPM -- although both groups had an elevated heart rate compared to healthy relaxed human subjects (there was no healthy control group). PVC's were also significantly more common in the non-survivors. Significant differences were also found in three laboratory tests: The survivors had a significantly lower peak troponin (0.03 vs 0.18); a significantly lower C reactive protein (97 vs 181); and a significantly lower innterleukin 6 value (30 vs 296). PVC's and non sustained ventricular tachycardia were infrequent with no difference between the groups. (This contrasts with some studies performed elsewhere). Atrial fibrillation occurred in 11.9% of patients and was not different between survivors and patients who died. There was one patient with asystole who died, one patient with pulseless electrical acitivity who died and one patient with ventricular tqcycaria who survived. It is pointed out that all of the life threatening arrhythmias were in patients with an elevated troponin.

Another distinguishing feature of the survivors is that they had a higher body mass index, possibly because the non-survivors ate or drank less. The difference was statistically significant, but not drastically different.

The discussion contains reasonable explanations for the results, and notes the limitations of the study. No correlation could be demonstrated between anti-inflammatory markers and ventricular arrhythmias because the serious ventricular arrhythmias were infrequent. There were also no effects of drug treatment on arrhythmias, but the sample size was too small to rule this out. Remdesevir was administered in this study, but to a small fraction of the patients and there this study was not designed to determine its effect on overall clinical outcome.

The statistical comparisons were limited to non-survivors vs. survivors in a group of patients who were hospitalized and on telemetry. It would have been interesting to obtain similar data on patients who tested positive but were not hospitalized, or even healthy controls. There is certainly data on atrial fibrillation, PVC's and non-sustained VT in patients who were studied before 2019.

It should also be mentioned that some patients with myocardial infaction have been avoiding hospitals because they fear exposure to covid 19, and that this has produced increased mortality. This should be mentioned with an appropriate reference. This paper does not conclusively establish why the patients who died had higher peak troponin levels, so it is possible that early hospital treatment could have been helpful. Also, as pointed out, there were no studies such as MRI or biopsy to confirm the presence of myocarditis.

Minor points: In figure 2 the three arrhythmias illustrated were presumably all obtained in different patients. This should be stated.

It could also be asked why the patients classified as DNR were kept on telemetry. Presumably patients sent to hospice are not on telemetry, but DNR patients are often sent non-acute wards with lower levels of nursing care.

6. PLOS authors have the option to publish the peer review history of their article (what does this mean?). If published, this will include your full peer review and any attached files.

Reviewer #1: No

---

## [Author Response · Author response to Decision Letter 0]

27 Nov 2020

We have addressed all the comments of the reviewer.

---

## [Decision Letter · Decision Letter 1]

14 Dec 2020

Cardiac Arrhythmias in Hospitalized Patients with COVID-19: A Prospective Observational Study in the Western United States

PONE-D-20-25283R1

Dear Dr. Cingolani,

We’re pleased to inform you that your manuscript has been judged scientifically suitable for publication and will be formally accepted for publication once it meets all outstanding technical requirements.

Kind regards,

Elena G. Tolkacheva, PhD

Academic Editor

PLOS ONE

Additional Editor Comments (optional):

Reviewers' comments:

Reviewer's Responses to Questions

**Comments to the Author**

1. If the authors have adequately addressed your comments raised in a previous round of review and you feel that this manuscript is now acceptable for publication, you may indicate that here to bypass the “Comments to the Author” section, enter your conflict of interest statement in the “Confidential to Editor” section, and submit your "Accept" recommendation.

Reviewer #1: All comments have been addressed

2. Is the manuscript technically sound, and do the data support the conclusions?

Reviewer #1: Yes

3. Has the statistical analysis been performed appropriately and rigorously? 

Reviewer #1: Yes

4. Have the authors made all data underlying the findings in their manuscript fully available?

Reviewer #1: Yes

5. Is the manuscript presented in an intelligible fashion and written in standard English?

Reviewer #1: Yes

6. Review Comments to the Author

Reviewer #1: (No Response)

7. PLOS authors have the option to publish the peer review history of their article (what does this mean?). If published, this will include your full peer review and any attached files.

Reviewer #1: No

---

## [Editor Report · Acceptance letter]

16 Dec 2020

PONE-D-20-25283R1 

Cardiac arrhythmias in hospitalized patients with COVID-19:A prospective observational study in the western United States 

Dear Dr. Cingolani:

I'm pleased to inform you that your manuscript has been deemed suitable for publication in PLOS ONE. Congratulations! Your manuscript is now with our production department. 

Kind regards, 

on behalf of

Dr. Elena G. Tolkacheva 

Academic Editor

PLOS ONE